# Relative Deprivation, Poverty, and Mortality in Japanese Older Adults: A Six-Year Follow-Up of the JAGES Cohort Survey

**DOI:** 10.3390/ijerph16020182

**Published:** 2019-01-10

**Authors:** Masashige Saito, Naoki Kondo, Takashi Oshio, Takahiro Tabuchi, Katsunori Kondo

**Affiliations:** 1Faculty of Social Welfare, Nihon Fukushi University, Aichi 470-3295, Japan; 2Center for Well-being and Society, Nihon Fukushi University, Aichi 460-0012, Japan; 3Department of Health Education and Health Sociology, The University of Tokyo, Tokyo 113-0033, Japan; naoki-kondo@umin.ac.jp; 4Institute of Economic Research, Hitotsubashi University, Tokyo 186-8603, Japan; oshio@ier.hit-u.ac.jp; 5Cancer Control Center, Osaka International Cancer Institute, Chuo-ku 541-8567, Osaka, Japan; tabuchitak@gmail.com; 6Center for Preventive Medicines, Chiba University, Chiba 260-0856, Japan; kkondo@chiba-u.jp; 7Center for Gerontology and Social Science, National Center for Geriatrics and Gerontology, Aichi 474-8511, Japan; 8The institute of Japan Agency for Gerontological Evaluation Study, Tokyo 110-0001, Japan

**Keywords:** relative deprivation, material poverty, relative poverty, mortality, older people

## Abstract

Most studies have evaluated poverty in terms of income status, but this approach cannot capture the diverse and complex aspects of poverty. To develop commodity-based relative deprivation indicators and evaluate their associations with mortality, we conducted a 6-year follow-up of participants in the Japan Gerontological Evaluation Study (JAGES), a population-based cohort of Japanese adults aged 65 and older. We analyzed mortality for 7614 respondents from 2010 to 2016. Cox regression models with multiple imputation were used to estimate hazard ratios (HRs) for mortality. Seven indicators were significantly associated with mortality: no refrigerator, no air conditioner, cut-off of essential services in the past year for economic reasons, and so on. Among participants, 12.0% met one item, and 3.3% met two items or more. The HRs after adjusting for relative poverty and some confounders were 1.71 (95%CI: 1.18–2.48) for relative deprivation, and 1.87 (95%CI: 1.14–3.09) for a combination of relative poverty and deprivation. Relative deprivation was attributable to around 27,000 premature deaths (2.3%) annually for the older Japanese. Measurement of relative deprivation among older adults might be worthwhile in public health as an important factor to address for healthy aging.

## 1. Introduction

Several studies have shown that relative poverty based on low income is significantly associated with poor health [1]. However, a relative income approach has limitations when attempting to capture the diverse and complex aspects of poverty. In order to assess poverty line, researchers of poverty have proposed the concept of relative deprivation to measure the lack of a living standard that most people in society enjoy. Townsend reported that people experience relative deprivation when they lack the resources to follow a proper diet, cannot participate in activities, and do not have living conditions and amenities that are customary, or are at least widely encouraged, in the societies in which they belong [2]. It has also been suggested that people who live in relative deprivation have different characteristics from those living in relative poverty [3,4,5].

The association between relative deprivation and mortality was examined by some ecological studies. The use of social indicator approaches such as the Townsend deprivation index or Carstairs deprivation score has found that people in relatively deprived areas have a higher risk for standardized mortality rates [6,7], cancer mortality rates [8], and suicide rates [9]. However, considering the possibility of ecological fallacy, analysis at an individual level is needed. Although some Japanese cross-sectional studies have shown an association with individual poor subjective health [10,11], to our knowledge, no study has examined the association between relative deprivation and premature death at the individual level.

Therefore, we examined the association between relative deprivation as one aspect of poverty and mortality among older Japanese adults after controlling for relative income poverty and other confounding factors.

## 2. Materials and Methods

### 2.1. Study Design and Participants

We used prospective cohort data from the Japan Gerontological Evaluation Study (JAGES), a large-scale population-based study of Japanese people aged 65 or older who were physically and cognitively independent. Baseline data were collected from August 2010 to January 2012, with a response rate of 66.3%. Of these, the present analyses used data on 7614 participants who answered a relative deprivation questionnaire, after excluding participants with missing information on sex and age. The average age of the respondents was 73.5 years (standard deviation (SD) = 5.6), and 53.0% were women. This study was performed on the basis of a collaborative research agreement with the municipality. Ethical permission (No. 13–14) was provided by the Ethics Board at Nihon Fukushi University.

### 2.2. Mortality Outcome

We retrieved information on death records from 2010 to 2016 from the government database of public long-term care insurance. This government database covers all respondents. Among these records, there were 514 (6.75%) deaths identified in the analysis sample.

### 2.3. Relative Deprivation and Relative Poverty

Commodity-based relative deprivation indicators have been developed using a consensual approach based on public opinion. These indicators were drawn from a review of daily necessities and basic needs in society [4,10,12,13,14,15,16]. Based on these previous papers, we evaluated 13 indices that equated with “lack of daily necessities,” “lack of living environment,” and “lack of social life” due to economic reasons. These factors are associated with a low standard of living in current Japanese society [5,11]. Lack of daily necessities indicators included having no television, no refrigerator, no air conditioner, no microwave oven, or no water heater due to economic reasons. Lack of living environment indicators included having no private toilet, kitchen, or bathroom in the house, and having a dining room that was not separate from the bedroom. Lack of social life indicators included having no telephone or ceremonial dress, being absent from family celebrations and events during the previous year due to economic reasons, and having essential services such as water, electricity, or gas cut-off in the previous year (except in cases of forgetting to make a payment). The relative deprivation index was assessed by counting the number of these items that the respondent experienced.

Relative poverty was defined as an income of less than half of the median annual equivalent income in the government statistics [17]; the threshold was 1.49 million Japanese yen in 2009. This definition is accepted by the Organisation for Economic Co-operation and Development (OECD), and is conceptually based on the relative approach of the Luxembourg Income Study [18]. We used annual pre-tax household income. For each response, we calculated the equivalent household income by dividing the income by the square root of the number of household members. This square root scale implies that, for instance, a household of four persons has needs twice as large as one composed of a single person (it is not quadrupled).

### 2.4. Covariates

Demographic variables included sex, age, years of education, and marital status at the baseline survey. In order to account for the health status at the baseline, presence of medical treatment, self-recognition of forgetfulness, and depressive symptoms were also considered. Medical treatment was determined by asking, “Are you currently receiving any medical treatment?”. Self-recognition of forgetfulness was measured by asking, “Do people around you notice your forgetfulness, for example, by telling you that you often ask the same thing?”. Depressive symptoms were assessed using the short version of the Geriatric Depression Scale (GDS-15), which was developed for self-administration in the community, using a simple yes/no format [19].

### 2.5. Statistical Analysis

First, we extracted the relative deprivation index that was associated with premature death among older Japanese adults by calculating crude hazard ratios (HRs) for mortality using Cox regression analysis. Second, we applied Cox regression analysis, starting with assessing the relationship between relative poverty (monetary poverty) and mortality, adjusting for the above covariates (Model 1). In Model 2, we assessed the association between relative deprivation and mortality, adjusting for relative poverty and covariates. The relative deprivation index was divided to ternary (none, only one, two and over) in Model 2. In addition, we examined the binary category (none or anyone) in Model 2a, and the quaternion category (none, one, two, three and over) in Model 2b, respectively. We also examined the combination of relative deprivation and poverty in Model 3. 

To mitigate potential biases caused by missing information in predictors and covariates, we adopted the multiple imputation approach, under the missing at random assumption. We generated 20 imputed data sets using the multiple imputation by chained equations procedure. Finally, we calculated population attributable risk percentage (PAR%) in an older Japanese population. This estimation assumed that the adjusted HRs truly reflected causal impact, and that our results represented the entire older Japanese population. Data on annual mortality were obtained from governmental reports [20]. We used STATA 15.1 for all analyses.

## 3. Results

The crude HR showed that the seven relative deprivation indicators were significantly associated with a higher risk for death, respectively (Table 1 & Appendix A). In particular, the experience of cut-off of essential services in the past year had a higher risk for premature death compared with other deprivation factors. A total of 15.3%, 3.3%, and 1.5% of respondents reported experiencing a single deprivation item, two or more deprivation items, and three or more deprivation items, respectively. A significantly greater HR for higher mortality was seen for these subjects, compared with non-deprived people: 1.53 (95% confidence interval (CI): 1.24–1.90), 2.10 (95%CI: 1.46–3.02), and 2.36 (95%CI: 1.41–3.95), respectively.

After adjustment for individual attributes and relative poverty (Table 2), among respondents with two or more deprivation items, mortality risk was 1.71 (95%CI: 1.18–2.48) times higher than that in non-deprived subjects (Model 2). Respondents with only one deprivation item did not have a significantly higher mortality risk. In addition, the adjusted HRs were 1.62 (95%CI: 0.97–2.69) for respondents with two items, and 1.82 (95%CI: 1.10–3.01) for those with three or more items (Appendix B, Table A1). Relative poverty had a marginally significant association with mortality. In addition, respondents who fell under both relative deprivation and relative poverty had 1.87 (95%CI: 1.14–3.09) times higher mortality risk compared with those who fell under non-deprivation and poverty. The HR for relative deprivation was comparatively higher than that for relative poverty (Model 3). When we analyzed raw data that did not impute missing values, the major results and trends were similar to those reported above (Appendix B, Table A2).

The estimation of PAR% showed that about 27,000 premature deaths (2.3% of all deaths) or about 15,000 premature deaths (1.2% of all deaths) could be avoided annually if there was less severe relative deprivation in Japan (Table 3).

## 4. Discussion

Relative deprivation is an important element in poverty, although it might be unsuitable for an international comparative study because the standard decent life varies by nation, culture, and period. To the best of our knowledge, this is the first study to examine the effect and impact of relative deprivation and relative poverty on mortality among older Japanese adults. Our results suggest the effectiveness of our seven relative deprivation indicators as social determinants of healthy ageing. Our findings are consistent with previous findings that analyzed poor social support [5,21] and subjective health [10,11]. Our results also showed that the association between the relative deprivation index and premature death remained even after adjusting for monetary poverty.

At the same time, our study added new evidence that relative deprivation has a stronger association with mortality than relative poverty if subjects experience relative deprivation in two or more items. There are several possible reasons for this finding. First, a relative deprivation index might capture severe poverty conditions better than a relative income approach. Whelan et al. revealed that people living in relatively deprived conditions experienced long-term and severe poverty throughout their life course [13]. Some studies also reported that people who have overlapping multidimensional disadvantages are more likely to be socially excluded, have poor self-rated health, and experience psychological distress [3,22]. Second, unlike monetary poverty, a poor standard of living (which relative deprivation measures) might be closely related to unhealthy lifestyles, including poor eating habits and nutrition, and lack of access to healthcare and welfare services. Third, relative deprivation might increase psychosomatic stresses and anxieties related to complaints or dissatisfaction with life [23,24]. Among Japanese older adults, a lower relative income compared with their reference group was associated with the onset of functional disability and death from cardiovascular diseases, regardless of the amount of the objective income [25,26].

Although the Europe 2020 strategy has adopted a strong material deprivation index as a goal for social inclusion in the next decade, there are few discussions regarding specific policy in Japan. Our results suggest that relative deprivation indicators could more accurately represent severe or absolute poverty in society that relative poverty indicators cannot address. The income approach has some limitations in the discussion of the poverty line, because the relationship between income and consumption behavior is complex. Relative deprivation indicators, which are composed of specific primary goods and resources, might be relevant to capture their part of “capabilities” [27]. Our results showed that subjects that fulfilled two or more deprivation indicators had a higher mortality risk, although there are some discussions about the cut-off point of relative deprivation indicators. The proportion of relative deprivation indicators was low, but PAR% was not. It is important to assess and discuss relative deprivation in addition to the conventional relative income approach. 

This study has several limitations. First, although we included indicators used in previous studies, our relative deprivation indicators did not cover the full range of daily resources among older people in Japan. Second, our analysis was limited to all-cause mortality. Third, our findings may be underestimated because people living in serious poverty and deprivation may have been less likely to participate in our survey. Finally, our data were not representative of the whole country. On the other hand, it is important to note that we did perform a large-scale survey concerning non-monetary poverty among older people in more than one municipality. Only a few studies have focused on relative deprivation among Asian older adults [5,11,28].

## 5. Conclusions

It is well-known that poverty is one of the social determinants of health. One important implication of our findings is that measurement of relative deprivation, along with relative poverty (monetary poverty), might be worthwhile in public health as an important factor for healthy aging. From a life course perspective, the impact of relative deprivation on health should be evaluated in older people as one of the cumulative disadvantages.

## Figures and Tables

**Table 1 ijerph-16-00182-t001:** Relative deprivation index—its prevalence and association with mortality.

		%	Mortality	
Item	Category		Crude HR	(95%CI)
No television	No	98.1	ref.	
	Yes (+)	1.9	1.83 *	(1.12–2.96)
No refrigerator	No	98.9	ref.	
	Yes (+)	1.1	2.01 *	(1.11–3.65)
No air conditioner	No	95.5	ref.	
	Yes (+)	4.5	1.51 *	(1.07–2.13)
No private bathroom	No	93.2	ref.	
	Yes (+)	6.8	1.45 *	(1.08–1.96)
No ceremonial dress	No	98.6	ref.	
	Yes (+)	1.4	1.84 *	(1.04–3.27)
Absence from family ceremonial occasions	No	94.6	ref.	
	Yes (+)	5.4	1.65 **	(1.21–2.26)
Cut-off of essential services in the past year	No	98.9	ref.	
	Yes (+)	1.1	2.12 *	(1.17–3.85)
Relative deprivation index ^a^	None	84.7	ref.	
	1+	15.3	1.53 ***	(1.24–1.90)
	None	84.7	ref.	
	1	12.0	1.38 *	(1.07–1.77)
	2+	3.3	2.10 ***	(1.46–3.02)
	None	84.7	ref.	
	1	12.0	1.38 *	(1.07–1.77)
	2	1.9	1.90 *	(1.16–3.14)
	3+	1.5	2.36 **	(1.41–3.95)

*** *p* < 0.001, ** *p* < 0.01, * *p* < 0.05. HR: hazard ratio, 95%CI: 95% confidence interval. (+) is related to relative deprivation. ^a^ This index was assessed by counting the number of items.

**Table 2 ijerph-16-00182-t002:** Hazard ratios (HRs) for association of mortality with relative deprivation in multiple-imputed dataset ^a,b^.

	Model 1	Model 2	Model 3
	HR (95%CI)	HR (95%CI)	HR (95%CI)
Relative poverty			
Non-poverty	ref.	ref.	
Poverty	1.26 * (1.02–1.56)	1.22 † (0.98–1.53)	
Relative deprivation ^c^			
Non-deprivation		ref.	
1		1.14 (0.87–1.49)	
2 +		1.71 ** (1.18–2.48)	
Combination ^d^			
No dep. & pov.			ref.
Poverty only			1.22 † (0.98–1.52)
Deprivation only			1.86 † (0.92–3.76)
Pov. & dep.			1.87 * (1.14–3.09)

** *p* < 0.01, * *p* < 0.05, † *p* < 0.10. HR: hazard ratio, 95%CI: 95% confidence interval. ^a^ Multiple imputation by chained equations was performed using relative deprivation index, relative poverty, sex, age, years of education, marital status, disease and/or impairment, self-recognition of forgetfulness, depressive symptoms (m = 20). ^b^ Sex, age, years of education, marital status, disease and/or impairment, self-recognition of forgetfulness, and depressive symptoms were controlled. ^c^ This index was assessed by counting the number of items. ^d^ Relative deprivation in combination variable was defined as respondents who fell under two and over deprivation index. Proportions of each category were as follows: No dep. & pov.: 70.1%; poverty only: 27.0%; deprivation only: 1.1%; and pov. & dep.: 1.7%. In addition, the proportion is not coincident with other tables, because it was confined to the respondents which answered the relative deprivation and poverty index.

**Table 3 ijerph-16-00182-t003:** Estimated population attributable risks (PARs) in Japan.

		Mortality		
	% Exposed ^a^	HR	PAR	
			% ^b^	n ^c^
Relative deprivation (1+)	15.3	1.25	3.7	44,197
(2+)	3.3	1.71	2.3	27,465
(3+)	1.5	1.82	1.2	14,577
Relative poverty	18.0	1.22	3.8	45,698

^a^ The % exposed of relative deprivation is in our study participants. That of relative poverty is from Japanese official statistics (comprehensive survey of living conditions). ^b^ PAR(%) = *Pe*(*HR* − 1)/(*Pe*(*HR* − 1) + 1); *Pe*: the proportion of exposure in the target population; *HR*: hazard ratio. ^c^ The denominator is the annual number of mortality among people 65 years and older in 2015 (N = 1,199,686) which was obtained from governmental reports.

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
