# Peer review of "Relative Deprivation, Poverty, and Mortality in Japanese Older Adults: A Six-Year Follow-Up of the JAGES Cohort Survey"

_ijerph, 2019, doi:10.3390/ijerph16020182_

Round 1

Reviewer 1 Report

This is a clear and well written paper even as English is not the first language of the authors. I think the findings are an important first step in examining relative deprivation and economic poverty among older Japanese adults and perhaps later, older adults elsewhere.

Author Response

We very much appreciate the comments made by reviewers. We have made the following specific changes to the manuscript in response to their helpful suggestions.

This is a clear and well written paper even as English is not the first language of the authors. I think the findings are an important first step in examining relative deprivation and economic poverty among older Japanese adults and perhaps later, older adults elsewhere.

Response;

We are very honoured to be able to receive positive comment.

Sincerely.

Reviewer 2 Report

This paper studied associations between mortality and relative deprivation/poverty in Japanese older adults. The paper is generally well-organized and written. However I do have following comments for the authors to consider:

Page 2 line 67-69: The authors should provide more details about the database of public long-term care insurance. Does the database covers all the JAGES participants or the 7614 participants who answered  relative deprivation questionnaire?

Page 2 line 89-90: Please provide a reference for the equivalent household income (why divided by the square root of the number of household members?)

Page 3 Line 102: What tests were used for crude hazard ratios? Chi-square tests? If so, that shouldn't be called hazard ratios. Please clarify.

Page 3 Line 104: Does "control variables" mean covariates in section 2.4?

Page 3 Line 106-107: please define binary, ternary and quaternion here.

Page 3 Line 108: What information is defined as "missing information"? predictors or covariates or outcome?

Page 3 Line 119-123: These two sentences need to be better expressed to eliminate confusions.

Page 4 Line 132: 'Relative poverty had a marginally significant association with mortality (p=0.066)" Which model are you referring to?

In terms of models (statistical analysis and Table 2), I would recommend to remove the models 2 and 4 and only report three models to convey the message..

It will be helpful to provide frequencies (percentages) of each cell of the combination of deprivation and poverty (model 5).

There were 13 items for the relative deprivation index and 7 of them (table 1) showed statistical significance. It seems later relative deprivation index was built on the 7 items. However, since the study was designed using 13 items, the authors might want to use all the 13 items, even though 6 items didn't show statistical significance in the crude analysis. When counting the total number of items (out of 13), these 6 items might still be meaning full. I would suggest the authors to do a secondary  analysis using all 13 items.

Author Response

We very much appreciate the comments made by reviewers. We have made the following specific changes to the manuscript in response to their helpful suggestions.

Page 2 line 67-69: The authors should provide more details about the database of public long-term care insurance. Does the database covers all the JAGES participants or the 7614 participants who answered relative deprivation questionnaire?

Response;

Thank you for your comment. The government database covers all the JAGES participants including the subjects of this paper. We added brief explanation in page 2 line 68.

2.2. Mortality Outcome:

We retrieved information on death records from 2010 to 2016 from the government database of public long-term care insurance. This government database covers all of respondents. Among these records, there were 514 (6.75%) deaths identified in the analysis sample.

Page 2 line 89-90: Please provide a reference for the equivalent household income (why divided by the square root of the number of household members?)

Response;

Recent OECD publications applied this square root scale which divides household income by the square root of household size. This implies that, for instance, a household of four persons has needs twice as large as one composed of a single person; it is not quadrupled at least. We added this explanation in page 2 line 90.

2.3. Relative Deprivation and Relative Poverty:

We used annual pre-tax household income. For each response, we calculated the equivalent household income by dividing income by the square root of the number of household members. This square root scale implies that, for instance, a household of four persons has needs twice as large as one composed of a single person; it is not quadrupled at least.

Page 3 Line 102: What tests were used for crude hazard ratios? Chi-square tests? If so, that shouldn't be called hazard ratios. Please clarify.

Response;

We estimated crude hazard ratios, applying Cox regression analysis. We added this point in page 3 line 104.

2.5. Statistical Analysis:

First, we extracted the relative deprivation index that was associated with premature death among Japanese older adults by calculating crude hazard ratios (HRs) for mortality using Cox regression analysis. Second, we applied Cox regression analysis, starting with assessing the relationship between relative poverty (monetary poverty) and mortality, adjusting for the above control variables (model 1).

Page 3 Line 104: Does "control variables" mean covariates in section 2.4?

Response;

In econometrics, the term "control variable" is usually used instead of "covariate". However, in order to avoid confusion, we modified to "covariates" in page 3.

2.5. Statistical Analysis:

Second, we applied Cox regression analysis, starting with assessing the relationship between relative poverty (monetary poverty) and mortality, adjusting for the above covariates (model 1). In models 2, we assessed the association between relative deprivation and mortality, adjusting for relative poverty and covariates.

Page 3 Line 106-107: please define binary, ternary and quaternion here.

Response;

We added informations about each categories as follows (page 3 line 108).

2.5. Statistical Analysis:

Relative deprivation index was divided to ternary (none, only one, two and over) in model 2. In addition, we examined binary category (none or anyone) in model 2a, and quaternion category (none, one, two, three and over) in model 2b, respectively.

Page 3 Line 108: What information is defined as "missing information"? predictors or covariates or outcome?

Response;

We added informations about multiple imputation (page 3 line 112).

2.5. Statistical Analysis:

To mitigate potential biases caused by missing information in predictors and covariates, we adopted the multiple imputation approach, under the missing at random assumption.

Page 3 Line 119-123: These two sentences need to be better expressed to eliminate confusions.

Response;

Thank you for your comment. In order to clarify the meaning of sentences, we modified these two sentences as follows (page 4 line 125).

3. Results:

The crude HR showed the seven relative deprivation indicators were significantly associated with a higher risk for death, respectively (Table 1). In particular, the experience of cutoff of essential services in the past year had a higher risk for premature death compared with other deprivation factors.

Page 4 Line 132: 'Relative poverty had a marginally significant association with mortality (p=0.066)" Which model are you referring to?

Response;

The estimation of "p=.066" was derived in model 2a (previous model 2). However, in order to avoid confusion, we eliminated this description.

3. Results:

Relative poverty had a marginally significant association with mortality.

In terms of models (statistical analysis and Table 2), I would recommend to remove the models 2 and 4 and only report three models to convey the message.

Response;

 Thank you for your very important suggestion. We removed the model 2 and 4 in table 2. We added one appendix which showed these two models. Then, we modified main text as follows.

2.5. Statistical Analysis (page 3 line 108):

Relative deprivation index was divided to ternary (none, only one, two and over) in model 2. In addition, we examined binary category (none or anyone) in model 2a, and quaternion category (none, one, two, three and over) in model 2b, respectively.

3. Results (page 3 line 132):

After adjustment for individual attributes and relative poverty (Table 2), among respondents with two or more deprivation items, mortality risk was 1.71 (95%CI: 1.18–2.48) times higher than that in non-deprived subjects (Model 2). Respondents with only one deprivation item did not have a significantly higher mortality risk. In addition, the adjusted HRs were 1.62 (95%CI: 0.97–2.69) for respondents with two items and 1.82 (95%CI: 1.10-3.01) for those with three or more items (Appendix A). Relative poverty had a marginally significant association with mortality. In addition, respondents who fell under both relative deprivation and relative poverty had 1.87 (95%CI: 1.14-3.09) times higher mortality risk compared with non-deprivation and poverty. The HR for relative deprivation was comparatively higher than that for relative poverty (Model 3). When we analyzed raw data that did not impute missing values, the major results and trends were similar to those reported above (Appendix B).

Appendix A (page 6 line 234):

Hazard Ratios (HRs) for association of mortality with other relative deprivation criteria in multiple-imputed dataset a,b

It will be helpful to provide frequencies (percentages) of each cell of the combination of deprivation and poverty (model 5).

Response;

Thank you for your comment. We revised footnote in table 2.

Table 2:

d Relative deprivation in combination variable was defined as respondent who fell under two and over deprivation index. Proportions of each categories were as follows: No dep. & pov.: 70.1%, poverty only: 27.0%. deprivation only: 1.1%, and pov. & dep.: 1.7%

There were 13 items for the relative deprivation index and 7 of them (table 1) showed statistical significance. It seems later relative deprivation index was built on the 7 items. However, since the study was designed using 13 items, the authors might want to use all the 13 items, even though 6 items didn't show statistical significance in the crude analysis. When counting the total number of items (out of 13), these 6 items might still be meaning full. I would suggest the authors to do a secondary analysis using all 13 items.

Response;

Thank you for important suggestion. Although we analyzed the total score using 13 items, it was not necessarily appropriate index for unmature death at the follow-up period. However, in order to present foundational information, we added prevalence and association with mortality in original thirteen relative deprivation index as supplementary material.

Supplementary Material: The following are available online at www.mdpi.com/xxx/s1, Table S1: Original thirteen relative deprivation index; its prevalence and association with mortality.

Again, we would like to thank the reviewers and editor again for their helpful suggestions. We believe that our paper is improved as a result of attending to their suggestions, and we hope that our paper is now acceptable for publication. We look forward to hearing from you.

Sincerely.

Reviewer 3 Report

This is an interesting study of how relative deprivation and poverty influences mortality in Japanese older adults.

Even though, as the author’s state, the results cannot be extrapolated, the methodology could be useful in order to obtain these data in other countries.

I think that it would be interesting if the authors could add information about if of the different items explored (no television, no refrigerator, etc) there is any of them that has more impact than the others. 

Author Response

We very much appreciate the comments made by reviewers. We have made the following specific changes to the manuscript in response to their helpful suggestions.

This is an interesting study of how relative deprivation and poverty influences mortality in Japanese older adults. Even though, as the author’s state, the results cannot be extrapolated, the methodology could be useful in order to obtain these data in other countries. I think that it would be interesting if the authors could add information about if of the different items explored (no television, no refrigerator, etc) there is any of them that has more impact than the others.

Response;

Thank you for your comment. Our original 13 relative deprivation indicators were based on several previous research. Unfortunately, we don't have other indicators concerning relative deprivation in this survey. As we had mentioned in strengths and limitations, we thought it is one of limitations in this paper. To describe distribution of these 13 items, we added supplementary material.

Supplementary Material: The following are available online at www.mdpi.com/xxx/s1, Table S1: Original thirteen relative deprivation index; its prevalence and association with mortality.

Again, we would like to thank the reviewers and editor again for their helpful suggestions. We believe that our paper is improved as a result of attending to their suggestions, and we hope that our paper is now acceptable for publication. We look forward to hearing from you.

Sincerely.

Round 2

Reviewer 2 Report

I appreciate the revision of the manuscript. I have one question still:

The author mentioned "A total of 15.3%, 3.3%, and 1.5% of respondents reported experiencing a single deprivation item, two or more deprivation items, and three or more deprivation items." This implies a total of 4.8% experiencing two or more deprivation items. However, the proportions of the combined categories were reported as "Relative deprivation in combination variable was defined as respondent who fell under two and over deprivation index. Proportions of each categories were as follows: No dep. & pov.: 70.1%, poverty only: 27.0%. deprivation only: 1.1%, and pov. & dep.: 1.7%" (Table 2 foot note). This gives a total of deprivation (two and above items) of 2.8%. 

Could authors double check on this?

Author Response

Response to reviewer 2:

We very much appreciate the comments made by reviewer. We have made the following specific changes to the manuscript in response to their helpful suggestions.

The author mentioned "A total of 15.3%, 3.3%, and 1.5% of respondents reported experiencing a single deprivation item, two or more deprivation items, and three or more deprivation items." This implies a total of 4.8% experiencing two or more deprivation items. However, the proportions of the combined categories were reported as "Relative deprivation in combination variable was defined as respondent who fell under two and over deprivation index. Proportions of each categories were as follows: No dep. & pov.: 70.1%, poverty only: 27.0%. deprivation only: 1.1%, and pov. & dep.: 1.7%" (Table 2 foot note). This gives a total of deprivation (two and above items) of 2.8%.  Could authors double check on this?

Response;

Thank you for your important comment. We rechecked our data. These descriptive statistics were based on the respondent who answered relative deprivation and poverty index. The unknown cases in relative poverty were excluded. Because the proportion of relative deprivation was higher in unknown cases in relative poverty, the proportion in the above became low (2.8%). However, there is no change in the proportion of relative deprivation (2+): 3.3% in total sample. In addition, your comment that 4.8% of them was experiencing two or more deprivation items is misunderstanding. It was just 3.3% as shown in table 1 and table 3. In order to avoid confusion, we modified to footnote as follows.

Table 2:

d Relative deprivation in combination variable was defined as respondent who fell under two and over deprivation index. Proportions of each categories were as follows: No dep. & pov.: 70.1%, poverty only: 27.0%. deprivation only: 1.1%, and pov. & dep.: 1.7%. In addition, the proportion is not coincident with other tables, because it was confined to the respondent which answered relative deprivation and poverty index.
